# Comparative Effects and Mechanisms of Chitosan and Its Derivatives on Hypercholesterolemia in High-Fat Diet-Fed Rats

**DOI:** 10.3390/ijms21010092

**Published:** 2019-12-21

**Authors:** Chen-Yuan Chiu, Tsai-En Yen, Shing-Hwa Liu, Meng-Tsan Chiang

**Affiliations:** 1Department of Botanicals, Medical and Pharmaceutical Industry Technology and Development Center, New Taipei City 248, Taiwan; cychiu@pitdc.org.tw; 2Department of Food Science, College of Life Science, National Taiwan Ocean University, Keelung 202, Taiwan; tsaienyen@gmail.com; 3Institute of Toxicology, College of Medicine, National Taiwan University, Taipei 100, Taiwan; 4Department of Pediatrics, College of Medicine and Hospital, National Taiwan University, Taipei 100, Taiwan; 5Department of Medical Research, China Medical University Hospital, China Medical University, Taichung 404, Taiwan

**Keywords:** high and low molecular weight chitosan, chitosan oligosaccharide, lipid metabolism

## Abstract

The present study investigated and compared the effects of different molecular weights of chitosan (high molecular weight chitosan (HC) and low molecular weight chitosan (LC)) and its derivatives (chitosan oligosaccharide (CO)) on cholesterol regulation in high-fat (HF) diet-fed rats. A diet supplementation of 5% HC, 5% LC, or 5% CO for 8 weeks showed hypocholesterolemic potential in HF diet-fed rats. Unexpectedly, a 5% CO-supplemented diet exerted hepatic damage, producing increased levels of aspartate aminotransferase (AST), alanine aminotransferase (ALT), and tumor necrosis factor-alpha (TNF-α). The supplementation of HC and LC, unlike CO, significantly decreased the hepatic total cholesterol (TC) levels and increased the fecal TC levels in HF diet-fed rats. The hepatic protein expression of the peroxisome proliferator-activated receptor-α (PPARα) in the HF diet-fed rats was markedly decreased, which could be significantly reversed by both HC and LC, but not CO, supplementation. Unlike the supplementation of CO, both HC and LC supplementation could effectively reverse the HF-inhibited/induced gene expressions of the low-density lipoprotein receptor (LDLR) and cholesterol 7α-hydroxylase (CYP7A1), respectively. The upregulated intestinal acyl-CoA cholesterol acyltransferase 2 (ACAT2) protein expression in HF diet-fed rats could be reversed by HC and LC, but not CO, supplementation. Taken together, a supplementation of 5% CO in HF diet-fed rats may exert liver damage via a higher hepatic cholesterol accumulation and a higher intestinal cholesterol uptake. Both HC and LC effectively ameliorated the hypercholesterolemia and regulated cholesterol homeostasis via the activation and inhibition of hepatic (AMPKα and PPARα) and intestinal (ACAT2) cholesterol-modulators, respectively, as well as the modulation of downstream signals (LDLR and CYP7A1).

## 1. Introduction

The hepatic manifestation, non-alcoholic fatty liver disease (NAFLD), has been an emerging public concern worldwide with an estimated 20%~30% prevalence among developing and developed countries, which is associated with elevated mortality and morbidity, particularly due to changing dietary habits in the consumption of western-style foods [1,2]. Hypercholesterolemia, a characteristic disorder of lipid metabolism that arose from overloads of cholesterol-, trans-fat-, and saturated fat-rich food, is a critical risk factor for the initiation and progression of NAFLD, leading to other chronic diseases, including cardiovascular diseases (CVD), metabolic syndrome, and cancer [3,4,5]. In addition, the increment levels of very-low-density lipoprotein cholesterol (VLDL-C), low-density lipoprotein cholesterol (LDL-C), and total cholesterol (TC) in serum or plasma have been manifested in the occurrence of hypercholesterolemia [6]. To date, the therapeutic remedy against hypercholesterolemia is commonly to abide in pharmacological medicines, including statins and fibrates, which may generate many adverse effects. Hence, exploring an alternative medicine, such as natural products, for controlling hypercholesterolemia has recently been an acceptable strategy against NAFLD.

Recently, a remarkable number of natural bioactive products have been explored from various marine organisms for the prevention or therapy of chronic diseases, including cardiovascular diseases, diabetes, arthritis, osteoporosis, neurodegenerative diseases, acquired immunodeficiency syndrome (AIDS), and cancers [7]. Chitosan, as well as the partial deacetylation of chitin composed of β-(1-4)-linked D-glucosamine and *N*-acetyl-d-glucosamine from the exoskeletons of marine crustaceans, including shrimps and crabs, has had its advantages manifested as the functional food (or nutraceutical) used in the prevention or treatment of the aforementioned chronic diseases, containing antibacterial, antidiabetic, antioxidant, anticancer, anti-inflammatory, and hypocholesterolemic properties [8,9,10,11]. Based on the characteristics of the biocompatibility of its structure and properties, there has recently been a growing interest in the biomedical and food applications of the depolymerized derivatives of high molecular weight chitosan (HC), low molecular weight chitosan (LC), and chitosan oligosaccharides (CO) against obesity, diabetes, and other metabolic disorders [12,13]. However, the comparative regulation of the lipid metabolism among HC, LC, and CO groups is not clear. Therefore, in the present study, high-fat (HF) diet-induced hypercholesterolemic rats were used as a metabolic imbalance animal model to interpret the comparative effects and mechanisms among HC, LC, and CO groups in regulating blood, hepatic, or fecal cholesterol levels.

## 2. Results and Discussion

### 2.1. Effects of High Molecular Weight Chitosan (HC), Low Molecular Weight Chitosan (LC), and Chitosan Oligosaccharide (CO) on Body Weight, Organ Weights, and Food Intake in High-Fat (HF) Diet-Fed Rats

Firstly, we examined the effects of HC, LC, and CO on body weight, organ weights, and food intake in HF diet-fed rats. As shown in Figure 1, rats fed with 5% LC and 5% CO for 8 weeks significantly reduced the final body weight compared to the HF group (Figure 1A). The food consumption was not significantly different among groups (Figure 1B). In addition, as shown in Figure 1C, liver weights are significantly increased in HF-fed rats, which could be reversed by the supplementation of either 5% HC or 5% LC; on the contrary, the supplementation of 5% CO in HF-fed rats has no significant improvement in the live weight. Moreover, there are no significant difference in intestinal weights among groups (Figure 1D). Picchi et al. (2011) have shown that HF diet-administered rats do not experience elevated weight gain but do develop a fatty liver compared to the rats that received the normal AIN-93 diet, which is similar to our result that the liver weight significantly increases without a significant increase in body weight gain in HF diet-fed rats [14]. Interestingly, the results show that the supplementation of 5% CO does not exhibit the therapeutic potential in HF diet-induced liver weight gains but causes a significant reduction in body weight, which was also found in diabetic Goto–Kakizaki rats with the supplementation of CO by Teodoro et al. (2016) [15], implying that a 5% CO supplementation may exert the liver damage potential.

### 2.2. Effects of HC, LC, and CO on Hepatic and Fecal Lipid Responses in HF Diet-Fed Rats

We next evaluated the effects of HC, LC, and CO on lipid profiles in the plasma, the liver, and in feces. As shown in Figure 2, the hypercholesterolemia is significantly influenced by increasing plasma total cholesterol levels (TC) (Figure 2A), LDL-C+VLDL-C (Figure 2B), and the TC/HDL-C ratio (Figure 2C) in rats fed with HF diets for 8 weeks. In addition, the elevated (TC, LDL-C+VLDL-C, and TC/HDL-C ratio) plasma lipid profiles in HF diet-fed rats are markedly attenuated by the supplementation of 5% HC, 5% LC, and 5% CO, suggesting that a diet supplemented with 5% chitosan and its derivatives effectively exerts the amelioration of the imbalance of circulated cholesterol and lipoprotein levels in HF diet-fed rats. Yao et al. (2008) have observed the hypocholesterolemic effect of 5% HC in streptozotocin-induced diabetic rats [16]. Moreover, Pan et al. (2016) have suggested that both chitosan and CO could lower circulated TC and LDL-C levels [17]. Therefore, these results are consistent with our present results that suggest that chitosan and its derivatives have significant therapeutic potential in hypercholesterolemia.

On the other hand, the supplementation of 5% HC and 5% LC shows a non-statistical inhibition in the plasma levels of biomarkers for liver damage, aspartate aminotransferase (AST) (Figure 3A), and alanine aminotransferase (ALT) (Figure 3B) in HF diet-fed rats; unexpectedly, the supplementation of 5% CO obviously elevated plasma AST and ALT levels compared to other groups, implying that liver injury, as well as the inflammatory response, was induced. The supplementation of 5% CO could also elevate the circulated levels of tumor necrosis factor-alpha (TNF-α) in HF diet-fed rats (Figure 3C). Kakino et al. (2018) have suggested that TNF-α plays a pivotal role in the development and progression of NAFLD by the association with lipid metabolism and inflammation in the liver [18], which is consistent with our finding in the present study that diets supplemented with 5% CO may induce liver damage. Moreover, as shown in Figure 4, rats fed with HF diets for 8 weeks exhibit a significant increase in hepatic (Figure 4A) and fecal (Figure 4B) TC levels. These abnormal TC levels, under the supplementation of 5% HC and 5% LC, were effectively ameliorated and promptly excreted in the feces. On the contrary, the supplementation of 5% CO has no significantly reversed effect on the abnormal TC levels in the liver and feces of HF diet-fed rats. The paradoxical phenomena under the supplementation of 5% CO in HF diet-fed rats, as mentioned above, may be based on the dosage of CO administration. A previous study described by Choi et al. (2012) has suggested that a 3% CO supplementation for 5 months could markedly improve the imbalance of circulated and hepatic lipid profiles [19]. Moreover, Yao et al. (2015) have indicated that a 3% CO diet could significantly reduce acetaminophen-induced hepatotoxicity [20]. Taken together, the therapeutic window of CO used against chronic diseases without observed adverse effects in the liver is a supplementation level of below 5%.

### 2.3. The Mechanism of HC, LC, and CO in the Livers and Intestines of HF Diet-Fed Rats

To determine the potential molecular mechanism of the preventive effects of HC, LC, and CO on hepatic and intestinal lipid profiles in HF diet-fed rats, we investigated the protein or gene expressions of lipometabolic modulators, including the adenosine monophosphate (AMP)-activated protein kinase-α (AMPKα), the peroxisome proliferator-activated receptor-α (PPARα), the LDL receptor (LDLR), Cholesterol 7α-hydroxylase (CYP7A1), and acyl-CoA cholesterol acyltransferase 2 (ACAT2). As shown in Figure 5A, the protein expression of phosphorylated AMPKα in the liver is shown to be a non-statistically significant inhibition in the livers of HF diet-fed rats. This could have exhibited a reversed trend if both 5% HC and 5% LC supplementations were used; similarly, HF diet-fed rats had a markedly inhibited hepatic protein expression of PPARα, which could be also reversed by both 5% HC and 5% LC supplementations (Figure 5B). Diets supplemented with 5% CO in HF diet-fed rats do not improve HF diet-inhibited protein expressions of phosphorylated AMPKα and PPARα (Figure 5A and 5B). Previous studies have indicated that AMPKα as well as PPARα activation plays pivotal roles in cholesterol homeostasis by inhibiting hepatic cholesterol synthesis and promoting the cholesterol efflux capacity, respectively [21,22]. Next, we respectively evaluated the upregulated/downregulated transcriptional expressions functioned for cholesterol synthesis, LDLR, and CYP7A1, while AMPKα was activated. As shown in Figure 5C and 5D, 5% HC and 5% LC can effectively reverse the HF-inhibited/induced gene expressions of LDLR and CYP7A1, respectively. Contrariwise, neither hepatic LDLR nor CYP7A1 gene expressions could be improved by the supplementation of 5% CO in HF diet-fed rats. Previous studies have suggested that LDLR may be responsible for lowering the synthesis of hepatic cholesterol by increasing the reabsorption of LDL-C [23,24]. Additionally, Bieghs et al. (2012) have indicated that LDLR gene knockout mice may develop prolonged hepatic inflammation and liver damage under long-term high-fat-high-cholesterol diet administration [25], which could be consistent with our result that 5% CO-supplemented diets could not improve abnormal levels of hepatic and fecal TC and also induced biomarkers (AST and ALT, Figure 3A,B) and the inflammatory modulator (TNF-α, Figure 3C) of liver injuries under the dysregulation of the AMPKα/LDLR pathway. On the other hand, Hoang et al. (2011) have shown that the efflux of cholesterol from the body is via conversion to bile acids [26], and CYP7A1 is notoriously functioned as the rate-limiting enzyme in the bile acid biosynthetic pathways [27]. Moreover, Ferrell et al. (2016) have suggested that CYP7A1-deficient mice could be against HF/high-cholesterol diet-induced metabolic disorders by increasing the fecal cholesterol efflux [28]. In this study, our findings show that a 5% CO supplementation cannot increase fecal cholesterol efflux (Figure 4B) due to the activation of CYP7A1 gene expression in HF diet-fed rats (Figure 5D). Furthermore, it is well-known that the absorption of cholesterol in the intestine is also a critical process in lipid homeostasis, and that ACAT2 is a key enzyme for modulating the intestinal absorption of cholesterol [29]. As shown in Figure 6, ACAT2 protein expression is upregulated in HF diet-fed rats, which can be reversed by the supplementation of 5% HC and 5% LC, but not 5% CO. Zhou et al. (2017) have discovered the cholesterol-lowering activity of the co-administration of berberine and evodiamine supplementation by the inhibition of ACAT2 protein expression [30], which is also consistent with our findings in the present study.

## 3. Materials and Methods

### 3.1. Materials

Low-MW chitosan and chitooligosaccharide from crustacean shells were purchased from Koyo Chemical Industry Company (Tokyo, Japan) and high-MW chitosan was purchased from Charming and Beauty Co, LTD (Taipei, Taiwan). Fourier transform infrared spectroscopy was used to detect the deacetylation degree of chitosan and chitooligosaccharide while high-performance liquid chromatography was applied to measure the average MW of chitosan and chitooligosaccharide. The average MW of high-MW chitosan, low-MW chitosan, and chitooligosaccharide was about 740 kDa, 80 kDa, and 719 Da, respectively. In addition, the deacetylation degree of high-MW chitosan, low-MW chitosan, and chitooligosaccharide were about 91%, 83.9%, and 100%, respectively.

### 3.2. Animals and Diets

Male six-week-old Sprague-Dawley (SD) rats were procured from BioLASCO Taiwan Co., Ltd. (Taipei, Taiwan) and were habituated at the animal facility of the National Taiwan Ocean University (Keelung, Taiwan) for one week of ad libitum feeding (Rodent Laboratory Chow, Ralston Purina, St. Louis, MO, USA). The animal husbandry conditions were as follows: in the temperature range 23 ± 1 °C, a light cycle of 12 h light/dark, and in the humidity range 40–60%. All animals were individually housed in stainless-steel-made cages. Acclimatized animals were randomly divided into five groups (*n* = 6 of each group): (1) standard rodent diet-fed rats (NC); (2) HF diet-fed rats; (3) HF diet-fed rats with 5% high-MW chitosan (HF + HC); (4) HF diet-fed rats with 5% low-MW chitosan (HF + LC); and (5) HF diet-fed rats with 5% chitooligosaccharide (HF + CO). In normal and high-fat diets, 5% cellulose was added in place of chitosan derivatives. The formulation of the experimental diets is shown in Table 1. The body weight was recorded weekly until sacrifice. After 8 weeks of experimental administration, the fasted animals were sacrificed by exsanguination during the inhalation of anesthesia. Plasma samples were collected for tests on liver function and indicators of hypercholesterolemia. The harvested liver and intestine tissues were shaved, defatted, weighed, flash-frozen, and stored at −80 °C until histological and biomolecular analyses were completed. Feces collected for 3 consecutive days before sacrifice were flash-frozen at −80 °C until fecal cholesterol content analysis was completed. All the animal protocols in this study obeyed the Guide for the Care and Use of Laboratory Animals (Institute of Laboratory Animal Resources, 2011) [31] and were ratified by the Animal House Management Committee of the National Taiwan Ocean University (approval number: 106029).

### 3.3. Determination of Total Cholesterol (TC) and Plasma Lipoprotein Cholesterol

The levels of TC in the plasma, liver, and feces were determined by the Audit Diagnostics Cholesterol assay kits (Cork, Ireland) according to the manufacturer’s protocols. Plasma lipoprotein cholesterol was separated and measured as previously described by Takehisa and Suzuki (1990) [32]. Briefly, plasma high-density lipoprotein cholesterol (HDL-C), low-density lipoprotein cholesterol (LDL-C), and very low-density lipoprotein cholesterol (VLDL-C) were segregated by density gradient ultracentrifugation (194000× *g* at 10 °C for 3 h) using a Hitachi SP85G preparative ultracentrifuge (Tokyo, Japan), and HDL-C, LDL-C, and VLDL-C were recovered by tube slicing.

### 3.4. Determination of Activities of Aspartate Aminotransferase (AST) and Alanine Aminotransferase (ALT)

The activities of AST and ALT were measured with RANDOX AST and ALT enzymatic kits (Antrim, UK), under an absorbance reading of 340 nm, equipped with a Hitachi U-2880A spectrophotometer (Tokyo, Japan) in a kinetic manner according to the manufacturer’s protocols.

### 3.5. Determination of Plasma Tumor Necrosis Factor-α (TNF-α)

The levels of plasma TNF-α were determined by an Assay Designs Rat enzyme-linked immunosorbent assay kit (Ann Arbor, MI, USA) according to the manufacturer’s protocols.

### 3.6. Western Blot Analysis

Protein extraction and Western blotting were assayed as described previously by Chiu et al. (2018) [33]. Briefly, the chilled radioimmunoprecipitation assay (RIPA) buffer with the Thermo Fisher Scientific cocktail of phosphatase and protease inhibitors (Waltham, MA, USA) was applied for tissue protein extraction. The Thermo Fisher Scientific BCA protein assay kit was used to determine the protein concentration of tissue samples. Next, 50–100 μg of tissue proteins were separated by 8–12% SDS-PAGE gel and then transferred onto polyvinylidene difluoride (PVDF) membranes (Bio-Rad, Hercules, CA, USA). After blocking in 5% non-fat dry milk (Fonterra Brands, Taipei, Taipei) or 3% bovine serum albumin (Sigma-Aldrich, St. Louis, MO, USA) solution for 1 h, PVDF membranes were probed with primary antibodies for anti-phosphorylated (Thr172) AMP-activated protein kinase α (p-AMPKα), anti-AMPKα (Cell Signaling Technology, Danvers, MA, USA), anti-PPARα, anti-Acetyl-CoA acetyltransferase (ACAT)-2, and anti-β-actin (Santa Cruz Biotechnology, Santa Cruz, CA, USA) at 4 °C overnight. The PVDF membranes were then probed with anti-rabbit or anti-mouse horseradish peroxidase-conjugated secondary antibodies (Cell Signaling Technology). Target protein expressions were detected by a BioRad Laboratories Enhanced Chemiluminescence kit (Redmond, WA, USA) and then exposed to a Fujifilm X-ray film (Tokyo, Japan). Image J 1.8 software (National Institutes of Health, Bethesda, MD, USA) was applied to quantify the band densitometry of target proteins.

### 3.7. Quantitative Reverse Transcription Polymerase Chain Reaction (qRT-PCR) Analysis

Total RNA extraction and qRT-PCR analysis were determined by using a TRIzol kit (Life Technologies, Carlsbad, CA, USA), an ABI StepOne™ Real-Time PCR system, and StepOne 2.1 software (Applied Biosystems, Foster City, CA, USA) as described previously by Chiu et al. (2017) [34]. Briefly, the real-time SYBR Green PCR reagent (Life Technologies, Carlsbad, CA, USA) was used to amplify the complementary DNA (cDNA), which was converted from the total RNAs (0.5–1 μg) extracted from liver tissues by avian myeloblastosis virus reverse transcriptase. The specific primers used are as follows: LDLR (forward: TGCTACTGGCCAAGGACAT; reverse: CTGGGTGGTCGGTACAGTG), CYP7A1 (forward: CTGCAACCTTTTGGAGCTTATT; reverse: GCACTCTGTAAAGCTCCACTC), and GAPDH (forward: CTGGAGAAACCTGCCAAGTATGAT; reverse: TTCTTACTCCTTGGAGGCCAGTA). An internal control GAPDH was used. The comparative threshold cycle method (ΔΔ*C*_t_) was used to determine the relative quantification of gene expressions. The levels of mRNA expressions were normalized by the GAPDH level. The fold changes of mRNA levels compared to the HF group were expressed as 2^−ΔΔ*C*t^.

### 3.8. Statistical Evaluation

All results are presented as the mean ± Standard Deviation (SD). The statistical significance (*p* < 0.05) is assessed by a one-way analysis of variance (ANOVA) with post-hoc Tukey’s test for multiple comparisons using GraphPad Prism V6.0 software (GraphPad Software, San Diego, CA, USA).

## 4. Conclusions

Both HC and LC can effectively ameliorate hypercholesterolemia and regulate cholesterol homeostasis via the activation and inhibition of hepatic (AMPKα and PPARα) and intestinal (ACAT2) cholesterol-modulators, respectively, as well as the modulation of downstream signals (LDLR and CYP7A1). Unexpectedly, 5% CO-supplemented diets do not improve a HF diet-induced imbalance of cholesterol levels in the liver and feces, and it may induce liver damage, suggesting that the therapeutic dosage of CO used against chronic diseases without observed adverse effects in the liver is a supplementation level of below 5%.

## Figures and Tables

**Figure 1 ijms-21-00092-f001:**
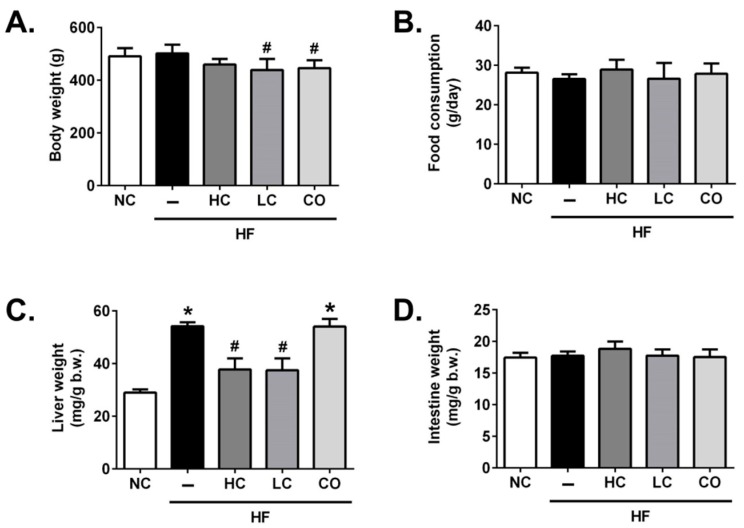
Effects of high-molecular weight and low-molecular weight (MW) chitosan (high molecular weight chitosan (HC), low molecular weight chitosan (LC), respectively) or chitosan oligosaccharide (CO) on body weight (**A**), food consumption (**B**), liver weight (**C**), and intestine weight (**D**) in high-fat (HF) diet-fed rats. The rats were fed with different experimental diets for 8 weeks. Data are presented as the mean ± SD (*n* = 6). * *p* < 0.05 as compared with the control group; # *p* < 0.05 as compared with the HF group. NC, normal control +5% cellulose; HF, high-fat diet +5% cellulose; HC, high-fat diet +5% high-MW chitosan; LC, high-fat diet +5% low-MW chitosan; CO, high-fat diet +5% chitosan oligosaccharide.

**Figure 2 ijms-21-00092-f002:**
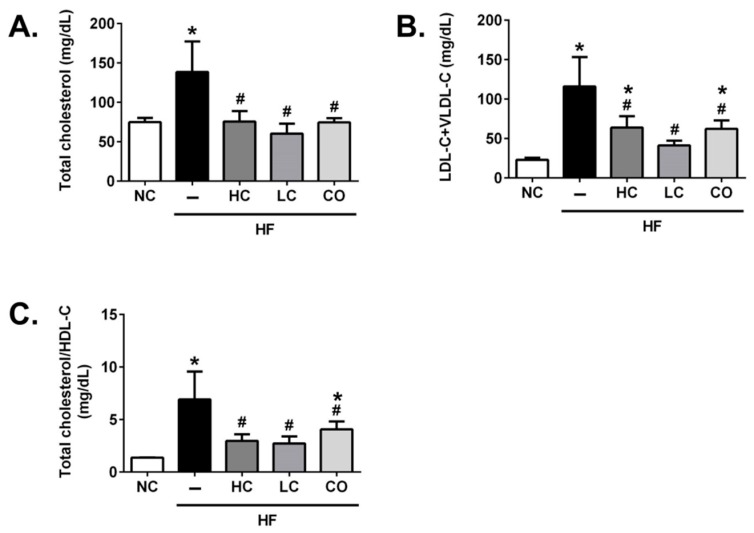
Effects of high-molecular weight and low-molecular weight (MW) chitosan or chitosan oligosaccharide on levels of total cholesterol (**A**), low-density lipoprotein cholesterol (LDL-C), very-low-density lipoprotein cholesterol (VLDL-C) (**B**), and the ratio of total cholesterol and high-density lipoprotein cholesterol (HDL-C) (**C**) in the plasma of high-fat (HF) diet-fed rats. The rats were fed with different experimental diets for 8 weeks. Data are presented as the mean ± SD (*n* = 6). * *p* < 0.05 as compared with the control group; # *p* < 0.05 as compared with the HF group. NC, normal control +5% cellulose; HF, high-fat diet +5% cellulose; HC, high-fat diet +5% high-MW chitosan; LC, high-fat diet +5% low-MW chitosan; CO, high-fat diet +5% chitosan oligosaccharide.

**Figure 3 ijms-21-00092-f003:**
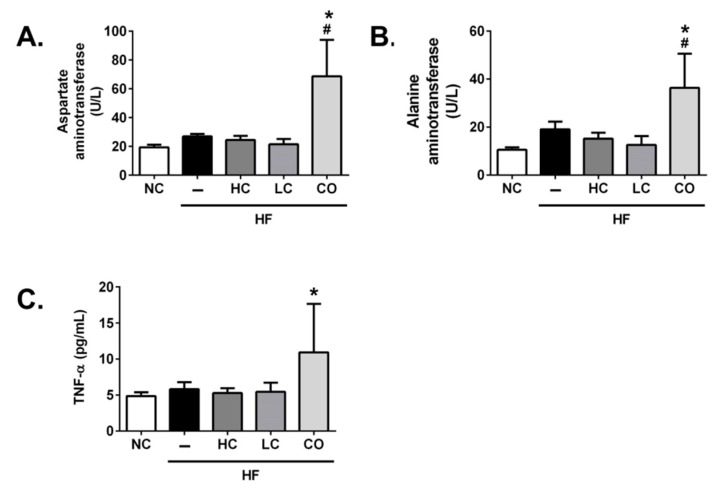
Effects of high-molecular weight and low-molecular weight (MW) chitosan or chitosan oligosaccharide on levels of aspartate aminotransferase (**A**), alanine aminotransferase, (**B**) and tumor necrosis factor-alpha (TNF-α) (**C**) in the plasma of high-fat (HF) diet-fed rats. The rats were fed with different experimental diets for 8 weeks. Data are presented as the mean ± SD (*n* = 6). * *p* < 0.05 as compared with the control group; # *p* < 0.05 as compared with the HF group. NC, normal control +5% cellulose; HF, high-fat diet +5% cellulose; HC, high-fat diet +5% high-MW chitosan; LC, high-fat diet +5% low-MW chitosan; CO: high-fat diet +5% chitosan oligosaccharide.

**Figure 4 ijms-21-00092-f004:**
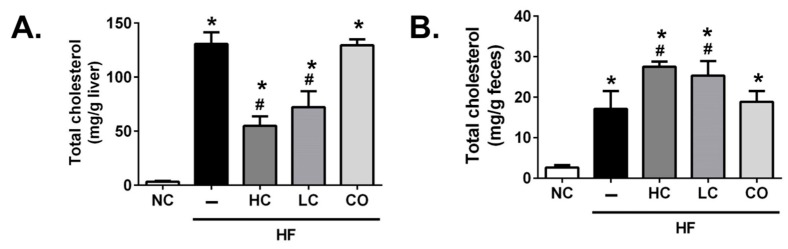
Effects of high molecular weight and low-molecular weight (MW) chitosan or chitosan oligosaccharide on levels of total cholesterol in the liver (**A**) and feces (**B**) of high-fat (HF) diet-fed rats. The rats were fed with different experimental diets for 8 weeks. Data are presented as the mean ± SD (*n* = 6). * *p* < 0.05 as compared with the control group; # *p* < 0.05 as compared with the HF group. NC, normal control +5% cellulose; HF, high-fat diet +5% cellulose; HC, high-fat diet +5% high-MW chitosan; LC, high-fat diet +5% low-MW chitosan; CO, high-fat diet +5% chitosan oligosaccharide.

**Figure 5 ijms-21-00092-f005:**
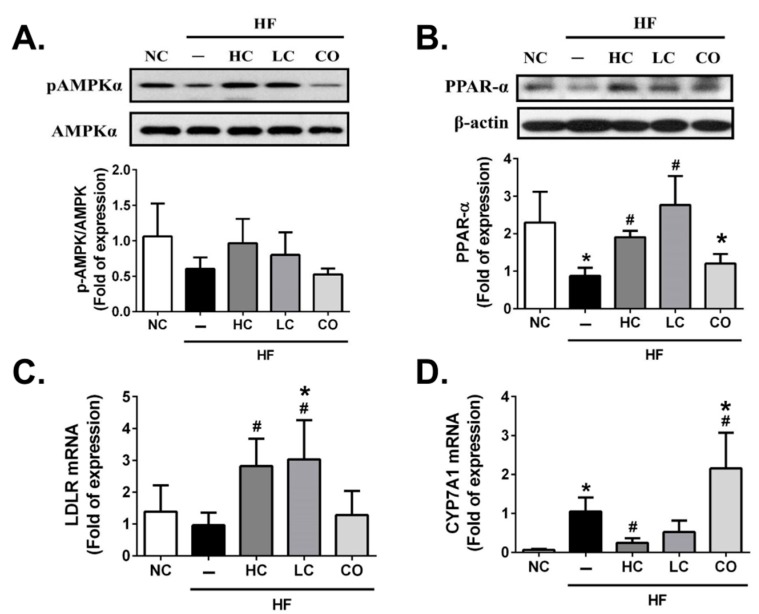
Effects of high-molecular weight and low-molecular weight (MW) chitosan or chitosan oligosaccharide on hepatic (AMP)-activated protein kinase-α (AMPKα) and peroxisome proliferator-activated receptor-α (PPARα) protein expressions and low-density lipoprotein receptor (LDLR) and cholesterol 7α-hydroxylase (CYP7A1) gene expressions in high-fat (HF) diet-fed rats. The rats were fed with different experimental diets for 8 weeks. Protein expressions of phosphorylated AMPKα/AMPKα (**A**) and PPARα (**B**) were determined using Western blot analysis. Gene expressions of LDLR (**C**) and CYP7A1 (**D**) were determined by a real-time reverse transcription polymerase chain reaction RT-PCR. Data are presented as the mean ± SD (*n* = 4–6). * *p* < 0.05 as compared with the control group; # *p* < 0.05 as compared with the HF group. NC, normal control +5% cellulose; HF, high-fat diet +5% cellulose; HC, high-fat diet +5% high-MW chitosan; LC, high-fat diet +5% low-MW chitosan; CO, high-fat diet +5% chitosan oligosaccharide.

**Figure 6 ijms-21-00092-f006:**
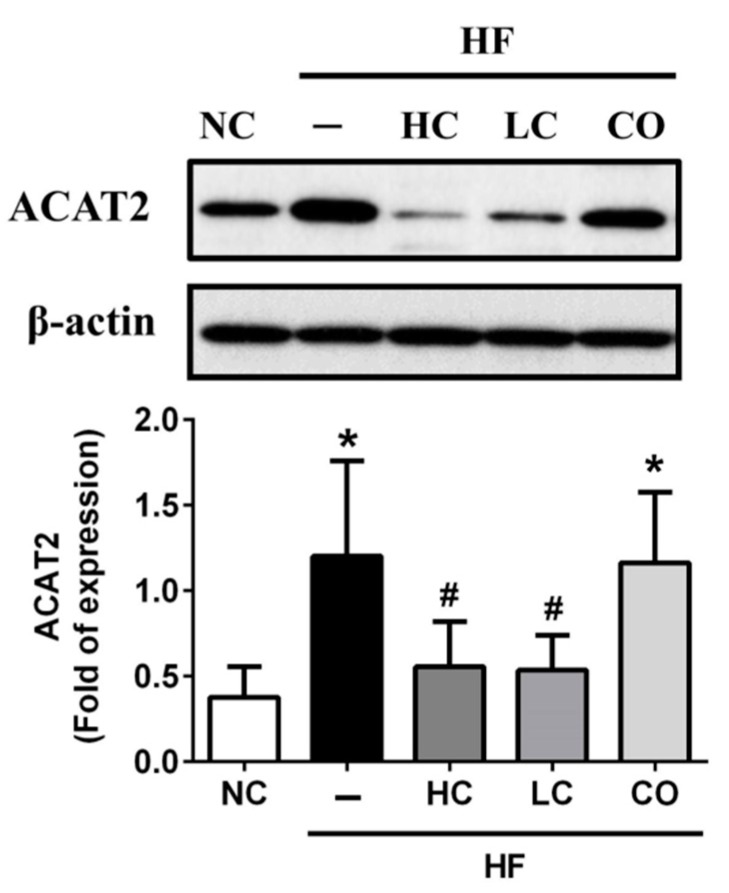
Effects of high-molecular weight and low-molecular weight (MW) chitosan or chitosan oligosaccharide on intestinal acyl-CoA cholesterol acyltransferase 2 (ACAT2) protein expression in high-fat (HF) diet-fed rats. The rats were fed with different experimental diets for 8 weeks. The protein expression of ACAT2 was determined using Western blot analysis. Data are presented as the mean ± SD (*n* = 6). * *p* < 0.05 as compared with the control group; # *p* < 0.05 as compared with the HF group. NC, normal control +5% cellulose; HF, high-fat diet +5% cellulose; HC, high-fat diet +5% high-MW chitosan; LC, high-fat diet +5% low-MW chitosan; CO, high-fat diet +5% chitosan oligosaccharide.

**Table 1 ijms-21-00092-t001:** Composition of experimental diets (%).

Ingredient (%)	NC	HF	HC	LC	CO
Corn starch	63.8	56.2	56.2	56.2	56.2
Casein	20	20	20	20	20
Lard	3	10	10	10	10
Soybean oil	2	2	2	2	2
Vitamin ^1^	1	1	1	1	1
Mineral ^2^	5	5	5	5	5
Cholesterol		0.5	0.5	0.5	0.5
Cholic acid		0.1	0.1	0.1	0.1
Choline chloride	0.2	0.2	0.2	0.2	0.2
Cellulose	5	5			
High-MW chitosan			5		
Low-MW chitosan				5	
Chitooligosaccharide					5
NC: Normal control diet					
HF: HF diet					
HC: HF diet + 5% high-MW chitosan					
LC: HF diet + 5% low-MW chitosan					
CO: HF diet + 5% chitooligosaccharide					
DD: Degrees of deacetylation					

^1^ AIN-93 vitamin mixture. ^2^ AIN-93 mineral mixture.

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
