# Peer review of "Comparative Effects and Mechanisms of Chitosan and Its Derivatives on Hypercholesterolemia in High-Fat Diet-Fed Rats"

_ijms, 2019, doi:10.3390/ijms21010092_

Round 1

Reviewer 1 Report

This manuscript reports the effects of chitosans of 3 different molecular weights on cholesterol metabolism in high fat diet (HF) fed rats. In results, high molecular weight chitosan (HC), low molecular weight chitosan (LC)) and chitosan oligosaccharide (CO) were effective to ameliorate HF-induced hepercholesterolemia but CO was associated with liver dysfunction and did not ameliorate accumulation of cholesterol in the liver. This report also checked protein and mRNA expressions of relevant genes and results showed that HC and LC but not CO ameliorated HF-induced changes. The methods seem proper and the results seem reasonable. But, there are some points to be improved as follows.
1. Results of post-hoc Tukey’s test was not described properly. In figure legend, only “Different letters indicate significant differences (p < 0.05).” was expressed but the meanings of letters (a, b and c) were not found. This may be expressed in the figure legend of Fig. 1 or somewhere in results and discussion section. In addition, meaningful differences are between normal control and HF or each chitosan group, between HF and each chitosan group and between each chitosan group. So, letter in NC group means little. So, the graphs may be redrawn accordingly.
2. 2. Fig. 4 is supposed to show total cholesterol in the liver (Fig. 4A) and that in the feces (Fig. 4B). However, as to Fig. 4B, the text and figure legends express “total cholesterol in the intestine”. In addition, “Liver” and “Feces” should be expressed in the graph.
3. In normal and high fat diet, 5% cellulose was added in place of chitosan derivatives. This is OK but origin of cellulose should be mentioned in the methods. Also, this should be mentioned in the text of 3.2 Animals and Diets.
4. The MW and DD of chitosan derivatives are repeatedly described in the text and the legend of Table 1. The latter may be omitted.
5. The expression, “the liver weights were significantly induced under the administration of HF diets” (lines 82-83) seems funny. Careful revision is needed.

Author Response

Reviewer-1

This manuscript reports the effects of chitosans of 3 different molecular weights on cholesterol metabolism in high fat diet (HF) fed rats. In results, high molecular weight chitosan (HC), low molecular weight chitosan (LC)) and chitosan oligosaccharide (CO) were effective to ameliorate HF-induced hepercholesterolemia but CO was associated with liver dysfunction and did not ameliorate accumulation of cholesterol in the liver. This report also checked protein and mRNA expressions of relevant genes and results showed that HC and LC but not CO ameliorated HF-induced changes. The methods seem proper and the results seem reasonable. But, there are some points to be improved as follows.

Results of post-hoc Tukey’s test was not described properly. In figure legend, only “Different letters indicate significant differences (p < 0.05).” was expressed but the meanings of letters (a, b and c) were not found. This may be expressed in the figure legend of Fig. 1 or somewhere in results and discussion section. In addition, meaningful differences are between normal control and HF or each chitosan group, between HF and each chitosan group and between each chitosan group. So, letter in NC group means little. So, the graphs may be redrawn accordingly.

Response: We appreciate the reviewer's comment. We have revised the description for statistical analysis in the text and figure legends of this revised manuscript according to the suggestion of reviewer.

 Fig. 4 is supposed to show total cholesterol in the liver (Fig. 4A) and that in the feces (Fig. 4B). However, as to Fig. 4B, the text and figure legends express “total cholesterol in the intestine”. In addition, “Liver” and “Feces” should be expressed in the graph.

Response: We appreciate the reviewer's comment. We have corrected this error and revised the Figure 4 of this revised manuscript according to the suggestion of reviewer.

In normal and high fat diet, 5% cellulose was added in place of chitosan derivatives. This is OK but origin of cellulose should be mentioned in the methods. Also, this should be mentioned in the text of 3.2 Animals and Diets.

Response: We appreciate the reviewer's comment. We have added the description about 5% cellulose in the methods section (3.2) of this revised manuscript according to the suggestion of reviewer.

The MW and DD of chitosan derivatives are repeatedly described in the text and the legend of Table 1. The latter may be omitted.

Response: We appreciate the reviewer's comment. We have omitted the description about MW and DD of chitosan derivatives in the Table 1 of this revised manuscript according to the suggestion of reviewer.

The expression, “the liver weights were significantly induced under the administration of HF diets” (lines 82-83) seems funny. Careful revision is needed.

Response: We appreciate the reviewer's comment. We have revised this sentence in this revised manuscript according to the suggestion of reviewer.

Reviewer 2 Report

The present study investigated and compared effects of high molecular weight chitosan, low molecular weight chitosan and chitooligosaccharide, purchased from a manufacturer, on cholesterol regulation in high-fat-diet-fed rats contributing to NAFLD.
In the introduction, the authors describe the reasons for the strategy of non-pharmacological treatment of NAFD and choosing chitosan.
To interpret the possible mechanisms of the effect of chitosan on lipid profile regulation, the authors used a wide range of methodologies such determination of lipid profile parameters, determination of liver enzymes activities, determination of TNF-a, and western blot analysis for protein and gene expression of lipometabolic modulators.
The authors chose an unconventional description of the Results and Discussion before the section Materials and Methods, which forces the reader to read the chapter once more after studying the materials and methodologies. It would be clearer to reorder these sections.
Based on the table no. 1, rats in the HF group received 5% cellulose, which in the chitosan diet groups were replaced with the appropriate chitosan. However, the description states that chitosan diets represented the HF diet with the addition of the appropriate chitosan. Please clarify, which option is the correct one.

Author Response

Reviewer-2

The present study investigated and compared effects of high molecular weight chitosan, low molecular weight chitosan and chitooligosaccharide, purchased from a manufacturer, on cholesterol regulation in high-fat-diet-fed rats contributing to NAFLD. In the introduction, the authors describe the reasons for the strategy of non-pharmacological treatment of NAFD and choosing chitosan. To interpret the possible mechanisms of the effect of chitosan on lipid profile regulation, the authors used a wide range of methodologies such determination of lipid profile parameters, determination of liver enzymes activities, determination of TNF-a, and western blot analysis for protein and gene expression of lipometabolic modulators.

The authors chose an unconventional description of the Results and Discussion before the section Materials and Methods, which forces the reader to read the chapter once more after studying the materials and methodologies. It would be clearer to reorder these sections.

Based on the table no. 1, rats in the HF group received 5% cellulose, which in the chitosan diet groups were replaced with the appropriate chitosan. However, the description states that chitosan diets represented the HF diet with the addition of the appropriate chitosan. Please clarify, which option is the correct one.

Response: We appreciate the reviewer's comment. Placed the Results and Discussion section before the section Materials and Methods is the standard format of the Journals that cannot be reordered by the authors. Moreover, we have revised the description for 5% cellulose in this revised manuscript according to the suggestion of reviewer.

Round 2

Reviewer 1 Report

 The meaning of the different letters (a, b and c) in the graphs are not clear yet. Probably, each letter has the meaning of "significant difference against NC or HF group" (results of Tukey's comparison). However, it is not mentioned at all. This should be mentioned in the first appearance of a, b and c. Or, if the letters mean differently in graphs, it should be mentioned in the graph legends. 

 It is not clear that NC values in the graphs have the letter "a". What does "a" mean? In these experiments, significant difference between NC and HF means "HF changed value from NC", difference between HF and either HC, LC or CO means "chitosan changed the effect of HF" and difference among HC, LC and CO means "different chitosan made different effects". So, it is not clear that NC has the letter "a" and again what do these letters mean?

Author Response

The meaning of the different letters (a, b and c) in the graphs are not clear yet. Probably, each letter has the meaning of "significant difference against NC or HF group" (results of Tukey's comparison). However, it is not mentioned at all. This should be mentioned in the first appearance of a, b and c. Or, if the letters mean differently in graphs, it should be mentioned in the graph legends. It is not clear that NC values in the graphs have the letter "a". What does "a" mean? In these experiments, significant difference between NC and HF means "HF changed value from NC", difference between HF and either HC, LC or CO means "chitosan changed the effect of HF" and difference among HC, LC and CO means "different chitosan made different effects". So, it is not clear that NC has the letter "a" and again what do these letters mean?

Response: We appreciate the reviewer's comment. We have made the graphs redrawn and corrected the statement of figure legends in the Results and Discussion section of this revised manuscript according to the suggestion of reviewer that *p < 0.05 as compared with control group and #p < 0.05 as compared with HF group were shown.